# Virulence and Immune Evasion Strategies of FMDV: Implications for Vaccine Design

**DOI:** 10.3390/vaccines12091071

**Published:** 2024-09-19

**Authors:** Gisselle N. Medina, Fayna Diaz San Segundo

**Affiliations:** 1National Bio and Agro-Defense Facility (NBAF), ARS, USDA, Manhattan, KS 66502, USA; 2Plum Island Animal Disease Center (PIADC), ARS, USDA, Orient Point, NY 11957, USA; 3National Institute of Health, NIAID, DMID, OBRRTR, Bethesda, MD 20892, USA

**Keywords:** foot-and-mouth disease, vaccines, livestock diseases

## Abstract

Foot-and-mouth disease (FMD) is globally recognized as a highly economically devastating and prioritized viral disease affecting livestock. Vaccination remains a crucial preventive measure against FMD. The improvement of current vaccine platforms could help control outbreaks, leading to the potential eradication of the disease. In this review, we describe the variances in virulence and immune responses among FMD-susceptible host species, specifically bovines and pigs, highlighting the details of host–pathogen interactions and their impact on the severity of the disease. This knowledge serves as an important foundation for translating our insights into the rational design of vaccines and countermeasure strategies, including the use of interferon as a biotherapeutic agent. Ultimately, in this review, we aim to bridge the gap between our understanding of FMDV biology and the practical approaches to control and potentially eradicate FMD.

## 1. Introduction

Foot-and-mouth disease (FMD), one of the most economically devastating diseases in the livestock industry worldwide, is caused by FMD virus (FMDV), a highly contagious, positive-sense, single-stranded RNA virus. FMDV belongs to the *Aphthovirus* genus of the *Picornaviridae* family. The FMDV viral genome is approximately 8500 nucleotides (nt) long and is organized in a single open reading frame (ORF) flanked by highly structured 5′ and 3′ non-coding regions (NCRs) (Figure 1). After infecting a cell, FMDV is rapidly translated into a single polyprotein, which is then processed into mature viral proteins. These proteins are essential for viral replication and enable the virus to spread quickly throughout the cell. The virus exhibits high antigenic variability, with six serotypes currently in circulation (A, O, Asia1, SAT1, SAT2, and SAT3), one “extinct” serotype C [1], and multiple circulating subtypes [2]. While clinical symptoms are similar among subtypes, cross-immunity and cross-protection are limited, potentially contributing to recurrent outbreaks in endemic regions. Interestingly, studies such as that by Grant et al. have provided insights into overcoming the challenges of cross-protection. Their research demonstrated that implementing a sequential vaccination approach with various FMDV serotypes can stimulate the production of antibodies that exhibit cross-reactivity with serotypes not directly encountered before the challenge [3]. Vaccination remains a crucial preventive and responsive measure against FMD. The improvement of current vaccine platforms could help control outbreaks in endemic areas and contribute to the potential eradication of FMD, reducing the economic burden on farmers and enhancing livestock productivity. However, it is crucial to highlight that FMDV maintains a natural reservoir in the wild, specifically in the African buffalo, thereby increasing the likelihood of transmission events between wildlife and cattle. To effectively address this issue, it is imperative to implement additional mitigation strategies, such as regulating movement, enhancing sanitary measures, and administering vaccines.

FMDV exhibits remarkable replication efficiency in susceptible hosts (i.e., swine and cattle), and exposure to the virus leads to nearly 100% morbidity. Complete disease manifestation occurs within 2–5 days after infection, resulting in fever, vesicular lesions, and ulcers in various areas, such as the mouth, tongue, nostrils, muzzle, feet, and teats [4]. FMDV can be transmitted through respiratory, oral, or percutaneous routes [5], which contributes to its virulence, as it can influence the number of infected animals, the ease of spread between hosts, and the efficiency of viral replication in the target tissues of the infected host [6]. This swift dynamic of infection depends on several factors, including but not limited to the presence of the cellular receptor, the intrinsic viral replication capacity, and the effectiveness of specific viral proteins to counteract the host immune response.

FMD viral virulence, as with other viruses, can be defined as “the ability to cause disease in an infected host” [7]. Considering the error-prone nature of viral RNA-dependent RNA polymerases, various genetically related FMDV variants (quasispecies) can emerge, each with different degrees of virulence [8]. This genomic variability can influence FMDV virulence across factors, such as the viral serotype/strain, host species, tissues, viral load, route of exposure, and the ability to evade the host’s immune system. Understanding these factors is critical for the development of vaccines to control FMD. In this review, we focus on the molecular features encoded in the FMDV genome and its evolutionary pathway for improving immune evasion strategies with the aim of enhancing the design and development of vaccines (Table 1).

## 2. Virulence Factors of FMDV and Their Roles in Immune Evasion

FMDV has evolved a cost-effective genome size strategy, referred to as “genomic economy” [61], allowing almost every region of its genome to fulfill specific functions within the virus life cycle. Various functions, including host cell entry, replication, and the blocking of host cell immune responses, are efficiently encoded in the compact genomic space contained within the icosahedron structure of FMDV. Recent in vitro studies have shown the diverse roles of FMDV non-coding regions (NCR), non-structural proteins (NSP), and structural proteins (SPs) in blocking immune responses at multiple levels [62,63]. This section analyzes the studies on viral determinants that shape the virulence of FMDV, highlighting how these research efforts have played a role in enhancing the development of more effective countermeasures against FMD. Refer to Figure 1 for a visual representation of the various regions of the FMDV genome discussed throughout this section.

### 2.1. Non-Coding Regions (NCRs)

The NCRs of FMDV including the 5′ terminal S (short) fragment, the internal ribosome entry site (IRES), and the 3′ NCR, house secondary RNA structures crucial for viral replication and translation. Various studies have highlighted that these non-coding segments can trigger an antiviral response when introduced independently into FMDV-susceptible cells and in an FMD mouse model [64]. This suggests that FMDV must employ additional mechanisms to evade the activation of antiviral responses during infection to ensure successful viral replication.

The initial RNA structure of the 5′ NCR is the S fragment, which encompasses approximately 350 nt, and adopts a stem–loop conformation. Investigations aimed at unraveling the 5′ NCR S fragment immunomodulatory functions have involved the introduction of deletions within the S fragment in the A24 cruzeiro (A24_Cru_) infectious clone. These manipulations yielded viable viruses displaying attenuating characteristics in vitro. Interestingly, a direct correlation was observed between the degree of the S fragment deletion in mutant viruses and their capacity to stimulate increased levels of IFN-β mRNA and various ISG mRNA levels within infected cells [57]. Importantly, the inoculation of mice with FMDV S fragment-deleted mutants confirmed in vivo attenuation but still allowed the production of antibody responses that effectively protected the animals when challenged with the wild-type (WT) virus. Most recently, deletions within the S fragment combined with an amino acid insertion in Lpro were detected in a naturally arising viral strain, which led to a shift in viral tropism [65]. Specifically, this virus only induced disease in pigs and not in cattle. These studies underscore both the plasticity of the S fragment in driving virus evolution that changes FMDV virulence and the potential for manipulating this region to create attenuated FMDV strains.

Downstream of the S fragment is the polycytidine (polyC) tract, recognized as a “viral security” RNA element, which spans between 150 and 250 nt in length, with greater variability in lab and vaccine strains (reviewed in [66]). Initial investigations into the potential virulence-related functions of these segments were carried out in vitro, focusing on evaluating the importance of the polyC tract length [21,67,68]. In these investigations, it was observed that deletions or truncations of the polyC segment did not have a detectable impact on virus viability [21]; however, increasing or decreasing this fragment significantly reduced its virulence in mice and/or cattle [67]. Although in vitro studies have cast doubt on the presence of a direct correlation between the length of the polyC tract and FMDV virulence, in vivo studies have suggested that increases in the polyC tract of FMDV are associated with persistent infection in the nasopharyngeal mucosa in cattle [69]. However, further studies confirming whether a long polyC tract offers an advantage for persistent FMDV remain to be evaluated.

The pseudoknots (PKs) of FMDV, integral to the 5’ NCR, typically occur in two to four tandem arrays, depending on the virus strain. Recent research has linked these PKs to the regulation of virus virulence; for instance, viruses with deletions in the PK region have shown an attenuated phenotype in bovine cells, but not in porcine cells, indicating a potential role for PKs in modulating viral host specificity [70]. Furthermore, viruses devoid of all PKs can still drive viral replication, although the production of infectious virions is compromised [71], suggesting a potential role of PKs in viral assembly due to the presence of a predicted packaging signal in this region [72]. These findings further highlight the multifaceted functions of FMDV pseudoknots in both replication and virion assembly.

The study of FMDV IRES virulence-related roles has proven challenging in in vivo contexts. However, through reverse genetics studies, the replacement of a single nucleotide (cytosine 351) within the FMDV IRES of serotypes A, Asia1, and O led to the creation of an attenuated virus that displayed immunoprotective features when assessed in mice, pigs, and cattle [73]. It is probable that alterations within specific IRES domains could impact the translation of the FMDV genome, driving the attenuation profiles of FMDV IRES mutants. This influence could compromise essential RNA–RNA interactions, as described by the interplay between FMDV IRES and 3′ UTR [74,75,76]. Moreover, these modifications could disrupt the interactions between FMDV IRES and cellular proteins [77,78,79].

### 2.2. Non-Structural Proteins (NSPs)

#### 2.2.1. Lpro

One of the most significant virulence factors identified in the FMDV genome is the leader (Lpro) coding region, which has been extensively studied through investigations involving deletions in this region in the context of the infectious clone [80]. FMDV Lpro is a papain-like protease (PLP) that effectively inhibits the cellular innate immune response, acting at both the transcriptional and translational levels. Through diverse mechanisms, Lpro effectively blocks the IFN response by targeting different cellular proteins (i.e., eIF4G, NF-kB, TBK, MDA5, ISG15, G3BP1/2, etc.) which have been described in detail elsewhere (reviewed in [81]). Thereby, modifications in the Lpro region, including full and partial deletions and/or amino acid substitutions, have proven to be a successful approach for creating attenuated strains with the potential to become effective modified live-attenuated (MLA) vaccines. For instance, the introduction of two specific amino acid mutations (I55A and L58A) in the SAP domain of Lpro in the A12 infectious clone (pRMC35) rendered a virus incapable of degrading NF-κB, leading to increased expression of various cytokines when tested in vitro [82]. Most interestingly, the in vivo assessment of this mutant virus yielded a remarkable outcome: pigs inoculated with the mutant virus remained completely healthy, without any sign of illness, viremia, or virus shedding, even when exposed to doses 100 times higher than those required to induce disease with wild-type (WT) virus [51]. Other mutations in Lpro (i.e., W105A and H138L) have resulted in somewhat similar attenuating profiles, mainly when evaluated in vitro, in FMD mouse models, and/or in swine [52,53]. However, a shared concern with all these mutants is the risk of reverting to a virulent state due to the few changes made in their genomes.

Attenuated FMDV leaderless strains in the A12 and A24_cru_ infectious clones have been studied for their virulence and evaluated in vaccine efficacy studies in pigs and cows [83,84]. These studies have shown promising results as potential vaccination platforms. Despite the reduced pathogenicity of the leaderless virus in swine and cattle [85], the modified virus was so attenuated that vaccinated animals were not completely protected against homologous wild-type (WT) virus challenge. However, the use of leaderless strains with other modifications, including the addition of restriction sites for capsid swapping and the incorporation of known DIVA (differentiating infected animals from vaccinated animals)-compatible antigen markers [23], is currently being developed as inactivated vaccine seeds to be formulated with adjuvants [24]. Together, these findings highlight the importance of Lpro to rationally design FMD vaccines.

#### 2.2.2. 2B

The coding sequence for the 2B protein of FMDV is highly conserved across strains [86], and recent studies have characterized 2B as a viroporin, meaning it can insert into cellular membranes and oligomerize to form an ion-conducting pore [87,88]. Recent studies have suggested that 2B can function in suppressing immune responses.

Weerawardhana et al. [89] demonstrated that FMDV 2B targets type I interferon (IFN) responses by degrading important cytosolic sensors, such as RIG-I and MDA5. These sensors are vital for detecting viral RNA and initiating subsequent immune responses. Furthermore, FMDV 2B has been shown to impact other host proteins, like NLRP3, NOD2, and LGP2 [90,91,92], thus contributing to different ways to modulate the cellular immune response during viral replication.

Limited evidence exists regarding the potential impact of alterations in the 2B region on FMDV virulence or infectivity during natural infection. However, a study conducted by Nishi et al. in 2019 [93] showed that on two isolated strains of FMDV (O/JPN/2000 and O/JPN/2010), which were associated with outbreaks of varying disease severity in cattle, three non-synonymous mutations specifically located in the 2B region were identified. Whether these mutations affect FMDV 2B’s ability to block the host’s innate immune response remains to be determined.

In the context of vaccine development via genetic engineering, the use of FMDV 2B has shown promise in optimizing vaccine design when administered via the Adenovirus type 5 platform. The incorporation of FMDV 2B was found to improve efficacy [31]. Furthermore, FMDV 2B of serotype O has been linked to the induction of protective CD4+ and CD8+ T cell responses [94].

#### 2.2.3. 2C

FMDV 2C, along with 2B, plays a crucial role in rearranging cell membranes during virus replication [95]. These rearrangements, also known as vesicularization, are essential for the formation of viral replication complexes during infection. FMDV 2C has been found to interact with distinct proteins, including vimentin and Beclin1, which are associated with cytoskeletal dynamics and autophagy processes, respectively [96,97]. Recently, in vitro experiments have shown that FMDV 2C can bind to a specific class of pattern recognition receptors (PRRs)—NOD2 [91]. The overexpression of FMDV 2C led to a significant decrease in NOD2 protein expression, highlighting its role in counteracting immune responses and its potential for immune evasion.

The virulence of FMDV may also be influenced by alterations in the coding sequence of the 2C protein. Importantly, certain substitutions in 2C, referred to as joker mutations, have been linked to enhanced viral fitness in diverse environments, including those influenced by antivirals like ribavirin and guanidinium chloride [98,99]. Supporting these observations, Yuan et al. [100] reported a unique substitution in 2C (T135I) that increased FMDV replication by stimulating viral RNA synthesis.

#### 2.2.4. 3A

The FMDV 3A protein plays a crucial role in determining host tropism [101,102], with its interactions with vimentin influencing membrane association [103] and its modulation of ER interactions regulating host protein secretion [104]. Interestingly, strains with specific codon deletions in the 3A coding region showed reduced virulence in cattle, while maintaining a virulent phenotype in pigs [102,105]. These deletions did not block the virus from initiating the infection of the nasopharyngeal mucosa, much like virulent FMDV strains [106]. Alterations in the 3A protein can confer new adaptive phenotypes to FMDV, offering the potential for enhancing vaccine safety during production. Recent studies have shed light on the role of FMDV 3A in immune responses, revealing interactions with DDX56 [107], as well as its impact on antiviral pathways triggered by IFN-β [108], RIG-I, and G3BP1 [109].

#### 2.2.5. 3B

FMDV 3B (also referred to as VPg) is critical for viral replication. Unlike other picornaviruses, the FMDV genome encodes three analogous copies—3B1, 3B2, and 3B3 [110]—that are conserved in almost all FMDV isolates [86]. Additionally, the 3B protein plays a crucial role in viral pathogenesis by suppressing the host immune response. This is achieved by the interaction and inhibition of host antiviral proteins such as RIG-I, resulting in the suppression of IFN-β and ISG expression [111].

Despite the observed conservation of all three 3Bs, genetic engineering has allowed the derivation of FMDV variants lacking one or featuring two non-functional 3B copies [112]. Interestingly, viruses lacking the first two 3B copies have exhibited a compromised ability to replicate in cell cultures and caused milder disease in pigs, suggesting that 3B may play a role in shaping the virus pathogenicity and host range [113]. Furthermore, mutations in FMDV 3B and 3D served as valuable negative antigenic markers in the development of a DIVA marker. This marker proved invaluable as a companion test for animals vaccinated with either an FMDV live attenuated vaccine candidate or inactivated vaccine lacking Lpro in the A24_cru_ backbone [23,24].

#### 2.2.6. 3C

FMDV 3C is a chymotrypsin-like cysteine protease required for the cleavage of the FMDV polyprotein into mature proteins. Additionally, it can cleave translation factors, specifically eIF4G and eIF4A [114], at later time points of infection, which can also contribute to the overall cellular shut-off of translation. Recently, the 3C protein has been demonstrated to impair the host’s innate immune responses during infection (reviewed in [63]). Most notably, 3C disrupts the JAK-STAT signaling pathway by preventing the STAT1/STAT2 nuclear translocation [115]; it can also induce the cleavage or degradation of NF-κB essential modulator (NEMO) [116] and cytosolic RNA sensors like MDA-5 and RIG-I [117] and modulate the phosphorylation of key signaling molecules in NF-κB signaling pathways and stress granule formation [118]. These functions were found to be dependent on the optimal protease activity of FMDV 3C, and any disruptions in 3C enzymatic activity, like substitutions in cysteine (i.e., C127T), can result in in vivo attenuation [117].

#### 2.2.7. 3D

FMDV 3D serves as the RNA-dependent RNA polymerase (RdRNA pol) and orchestrates the synthesis of a complementary RNA strand using the viral RNA as a template [62]. Given its central role in viral replication, FMDV 3D has emerged as a key target for antiviral drug development.

Since FMDV 3D polymerase lacks a proofreading function, high mutation rates in its viral genome are expected. Indeed, this contributes to the rapid evolution of the virus, enabling it to overcome bottlenecks during intra-host transmission and adapt to fluctuating environments [119]. Therefore, modulation of 3D polymerase’s fidelity can be employed as a strategy to attenuate FMDV replication. Several studies have indeed demonstrated that both natural variations and targeted modifications of specific amino acids within the 3D protein can influence FMDV pathogenicity [55,93,120,121]. Notably, changes in 3D that result in altered RdRpol fidelity have been shown to attenuate viral replication in FMDV animal models [55]. This attenuation is a sought-after characteristic for the generation of safer and genetically stable vaccine strains. However, the following question arises: can we solely rely on the use of 3D fidelity variants for the development of FMDV vaccine candidates? While high-fidelity variants may hold promise due to their potential to minimize mutation rates [122], thereby reducing the likelihood of virulence, it may be prudent to integrate these with additional, established attenuating mutations to synergistically diminish the risk of reverting to a virulent FMDV.

### 2.3. Structural Proteins (SPs)

The structural proteins of FMDV play a critical role in its life cycle, including genome packaging, the assembly of viral particles, and entry into host cells to initiate infection. The P1 precursor protein is composed of all the necessary sequences for viral capsid formation. These capsids display a marked vulnerability to acidic conditions and elevated temperatures, which can cause them to disassemble [17,123]. Importantly, the degree of stability can vary among serotypes, with FMDV O, and SAT being the most unstable [124,125]. This is a consideration that is essential for the production of stable virus-like particles when developing them as a vaccine platform.

The most notable characteristic of the FMDV viral capsid is the G-H loop, located in residues 140–160 of the VP1, which contains a highly conserved arginine-glycine-aspartate (RGD) motif [126]. The VP1 G-H loop is a highly antigenic determinant of the virus, with synthetic peptides mimicking this sequence known to induce protective neutralizing antibody responses [127,128]. Additionally, the VP1 G-H loop is critical for the attachment to integrin receptors on host cells, which are essential for FMDV entry [129]. The dual function of this loop facilitates the evolution of the FMDV antigenicity and its specificity for host cells, a feature that significantly contributes to its ability to adapt to diverse selective pressures.

In addition to providing the basis for antigenic variation to escape antibody neutralization, FMDV structural proteins have also been implicated in the suppression of immune responses, at least in vitro. In a study by Zhang et al. [130], it was discovered that the structural protein VP1 interacts with the cellular kinase TLP2, prompting the ubiquitination and subsequent degradation of TLP2, which leads to the diminished expression of antiviral cytokines. Similarly, research by Li et al. (2016) demonstrated that VP3 interacts with the cellular adaptor protein VISA, blocking the IFN-β signaling pathway [131]. These recent findings highlight the multifaceted role of FMDV structural proteins, which not only confer potent antigenicity and facilitate the evasion of antibody neutralization but also play a crucial role in modulating immune responses, particularly those driven by the innate immune system.

## 3. Host Pathogenesis and Immune Response to Understand FMDV in In Vivo Immune Evasion Strategies

FMDV has a wide host range, with each animal species exhibiting slightly different variations in disease pathogenesis. Understanding the pathogenesis from the point of initial infection to the triggered local and systemic immune responses—both innate and adaptive—is key to grasping how the virus impacts various tissues. A deeper understanding of how FMDV evades immune defenses in economically relevant species, such as cattle and swine, is crucial. This knowledge is essential for the rational development of vaccine and control strategies, which are necessary for managing outbreaks and reducing the burden in endemic regions.

## 4. Pathogenesis

Although the gold standard for a successful FMD vaccine would be to use a universal vaccine effective across all species, marked differences in the pathogenesis among cattle and swine may require tailored vaccines. In cattle, the initial FMDV infection targets the lymphoid-associated epithelium of the nasopharyngeal mucosa [6,132], followed by a clinical phase in which the virus spreads systemically and multiplies in vesicular lesions at peripheral sites, including the oral mucosa and coronary bands of the feet [132]. By contrast, primary FMDV infection in pigs originates in the epithelial crypts of the oropharyngeal tonsils [133]. Although the initial site of infection differs anatomically between cattle and pigs, the micro-anatomic and phenotypic characteristics of the infected lymphoid-associated epithelium are strikingly similar in both host species; therefore, a vaccine that effectively induces robust mucosal immunity could provide early protection universally. A significant difference between cattle and swine is the carrier state—approximately 50% of cattle may harbor a prolonged subclinical persistent phase of infection, a phenomenon not observed in pigs [134,135]. During this carrier state, infectious FMDV remains confined to distinct regions of the lymphoid-associated epithelium of the nasopharyngeal mucosa, similar to the primary infection sites [134,136,137]. Targeting these mucosal sites could not only reduce transmission but also prevent long-term persistence in cattle. This approach mirrors strategies against other respiratory viruses like SARS-CoV2, in which vaccine formulations are being developed to stimulate local mucosal immunity, particularly in the nasopharyngeal tract, to limit viral replication and transmission [138].

## 5. Innate Immune Responses In Vivo

### 5.1. Cytokine Immune Response

Viral infections trigger the innate immune system, serving as the first line of defense, which involves various cytokines, including the production of interferons (IFNs). At the onset of FMDV infection in cattle, studies have shown mixed results regarding the local antiviral responses in the nasopharyngeal mucosa; some have indicated minimal changes in IFN expression [6,106,132,139,140]. Systemically, cattle exhibit a robust activation of type I and III IFNs during the clinical phase of FMD [139,140,141,142,143], along with increased expression of several immune-related genes (i.e., Mx-1, OAS-1, CXCL10, ISG15, OAS1, and RIG-I) in PBMCs [144]. By contrast, the systemic IFN response in swine varies with the FMDV serotype [145,146,147] and, in general, remains less understood. Moreover, FMDV infection influences other cytokines in both swine and cattle. The early phases of infection show a systemic decrease in pro-inflammatory cytokines (IL-1β, IL-6, and TNFα), accompanied by an increase in the anti-inflammatory cytokine IL-10 [51,148,149]. This shift most likely impacts the cellular immune response. Given that FMDV can block the IFN signaling pathway to replicate effectively, the prophylactic use of IFNs or other cytokines constitutes a possible route to rapidly block the virus after infection, and IFNs could also be formulated with FMD vaccines as adjuvants (reviewed in [150]).

### 5.2. Cellular Immune Response

Following the initial FMDV infection, the virus encounters different cells of the innate immune system, including NK cells and antigen-presenting cells (APCs). This interaction occurs through the phagocytosis of infected epithelial cells [151] and/or direct contact with damaged infected tissue [152]. Additionally, macrophages (Mφ) [153] and dendritic cells (DCs) can internalize FMDV via antibody complexes [154]. Understanding how FMDV affects these innate immune cells is critical for enhancing vaccine designs by improving early immune responses, innate to adaptive immunological transitions, antigen presentation, and cytokine environment.

Natural killer (NK) cells: In swine, NK cells decline after infection, possibly due to reduced cytokine levels (IL-2, IL-15, and IL-1 [155]). Conversely, bovine NK cells show enhanced cytotoxic activity against bovine epithelial cells in vitro [156]. Recent studies have highlighted the importance of NK cells in orchestrating a diverse immune response and facilitating B-cell interactions for pathogen-specific immunity [157]. Although early therapeutic targeting of NK cells has shown promise in viral infections like vaccinia viral infection [158], its application in vaccine strategies for FMDV and other livestock viruses remains unexplored.

Dendritic cells (DCs): DCs are critical in initiating primary immune responses, acting as a bridge between innate and adaptive immunity. These cells can be broadly classified into two main types: plasmacytoid DCs (pDCs), which specialize in the production of cytokines, most importantly type I and III IFNs, and conventional DCs (cDCs), known for being potent APCs [159]. In the clinical phase of FMDV infection, both cattle and swine cDCs are stimulated by the virus to produce IL-10, skewing the immune response towards a humoral rather than a T-cell mediated adaptive response [160,161], even when the FMDV infection of these cells is mostly abortive [154,160]. Furthermore, FMDV blocks the maturation of porcine DCs into cDCs [160], reducing their response to TLR ligands [162], and in the case of cattle, FMDV infection downregulates MHC-II expression and impairs exogenous antigen processing in all bovine cDC populations [149]. The impact of FMDV on pDCs varies between cattle and swine. In cattle, there was an increase in mature CD4+ MHC-II+ pDCs during FMDV infection, while immature CD4+ MHC-II- pDCs levels declined [149]. Conversely, swine experienced a depletion in pDCs in the peripheral blood, with the remaining cells producing less IFNα upon ex vivo stimulation [146]. Tissue-resident Langerhans cells (LCs), a distinct subset of DCs located in the epidermis, are also affected by FMDV infection. Although FMDV infection of LCs is abortive [163], it still impairs their ability to produce IFNα after ex vivo stimulation in infected swine [162]. In order to enhance the efficacy of vaccines, strategies that expedite the activation of DCs, especially resident DCs in mucosal tissues can be considered. Incorporating adjuvants that target the activation pathways of DCs—mainly through the induction/expression of cytokines and chemokines that regulate DC trafficking, such as MIP-3α [164] and Flt3L [165]—could significantly improve vaccine performance.

Follicular DCs (FDCs), which reside in the lymphoid follicles of nearly all secondary lymphoid organs, are specialized in capturing and preserving antigens with their dendritic extensions [166]. Research involving cattle has indicated that FMDV can remain bound to FDCs within the lymphoid tissues even after the acute phase of infection has subsided [167]. Recent studies [168] have revealed that the proteins CR2 and CR1 play a role in this binding process, potentially influencing the retention of FMDV antigens on FDCs and the production of high-avidity and neutralizing antibodies. Further understanding of these interactions could lead to a more targeted approach in vaccine design.

Monocytes/Mφ: these cells are essential for the rapid clearance of viruses at infection sites. In the case of FMDV, monocytes/Mφ are susceptible to infection via antibody-dependent internalization. Although Mφ FMDV infection is abortive, the virus remains infectious within these cells for up to 24 h, suggesting that Mφ may act as disseminators of viable virions to distant sites of the body [151]. Studies on mouse models of FMDV [169] in which FMDV undergoes similar non-progressive replications in peritoneal Mφ, as in swine [170], have shown that depleting Mφ in vivo after vaccination drastically diminishes protection against FMDV challenges [171]. This underscores the importance of the Mφ population in early protective immune responses to FMDV.

Overall, the evidence indicates that FMDV significantly influences different aspects of the innate immune response, displaying unique patterns across different natural host species, which helps make the virus a successful pathogen. Although further research is warranted, potential strategies are being considered and could be developed to counteract the virus effect and induce a better response of the innate immune system.

## 6. Adaptive Immune Responses In Vivo

### 6.1. Humoral Immune Response

The humoral immune response plays a critical role in overcoming FMDV infection, with systemic neutralizing antibodies serving as key mediators for clearing the virus [172]. The rapid activation of B cells leads to early antibody response, as evidenced by detectable serum IgM within 3–4 days after infection in cattle, followed by peaks of IgA and then IgG within the following 1–2 weeks [173,174,175]. This response is not only systemic but also local, with antibodies detected in the oronasal and esophageal–pharyngeal regions as early as 4 days after aerosol exposure [176]. On the other hand, the local mucosal humoral immunity in swine is not as well characterized.

Increasing B-c-ell maturation and survival, along with promoting rapid isotype switching, could enhance vaccine efficacy and prevent carrier states in animals. Molecules such as B-cell-activating factor (BAFF) and a proliferation-inducing ligand (APRIL) are known to promote long-term B cell survival and maintain IgA and IgG responses [177,178]. Additionally, CD40 ligation enhances the expression of receptors that bind BAFF and APRIL [179], contributing to B-cell survival and function. These factors have shown promise as mucosal adjuvants against several infectious diseases. For instance, influenza and HIV vaccines incorporating some of these molecules in their design have elicited higher antibody titers and broader protection [180,181]. In livestock, Ad5 vectors expressing CD40L in combination with Porcine Reproductive and Respiratory Syndrome virus (PRRSV) antigens significantly reduce viremia in pigs [182]. Therefore, the coadministration of these molecules with existing FMD vaccines could amplify the humoral response, particularly at mucosal sites, potentially overcoming the challenge of the carrier status among cattle and other ruminants.

### 6.2. Cellular Immune Response

FMDV infection can also impact T-cell dynamics, affecting both alpha-beta (αβ) and gamma-delta (γδ) T-cell lineages. Early during FMDV infection, there is a transient lymphopenia in both cattle and swine, with an impairment in αβ T cell function, persisting beyond the initial lymphopenia phase [160,183]. Contrary to earlier beliefs, apoptosis is not the cause of this lymphopenia [183,184]; instead, elevated IL-10 levels during infection may be implicated [149,160]. Recent studies using IL-10 knockout mice have shown that blocking the IL-10 signaling pathway can prevent lymphopenia by reducing apoptosis and altering lymphocyte trafficking and co-inhibitory expression, thereby enhancing survival in FMDV-infected mice [185]. On the other hand, the response of γδ T cells during FMDV infection is slightly different. In cattle, WC1^+^ γδ T cells exhibit an activated phenotype, with changes in surface markers and increased IFNγ expression after FMDV infection [155], potentially acting as APCs [186]. However, more studies are needed to determine whether the changes observed in this cell population are the result of direct virus–cell interactions or a bystander consequence.

Despite species-specific responses to FMDV, the virus generally induces an immunosuppressive environment favoring a Th2 cell/cytokine-like environment that induces a strong neutralizing antibody response. Enhancing a T-cell response through vaccination strategies could thus improve protection. Importantly, highly interspecies MHC-restricted Th lymphocyte epitopes have been identified within the FMDV genome, such as the VP4 structural protein segment [187], 3D, and 3A [187,188]. Vaccine research is actively exploring platforms that incorporate these epitopes or employ adjuvants to enhance T-cell-mediated immune responses. Some of these will be discussed in depth in Section 4.

## 7. Vaccine Design Strategies against FMDV

While some parts of the world are free from FMD, FMDV remains endemic throughout much of Africa and Southern Asia, where six of the seven identified serotypes persistently circulate [189]. In these areas, the primary control measure against the disease is the use of approved vaccines based on chemically inactivated FMDV strains. Meanwhile, South American regions have witnessed significant advancements in controlling FMD through diligent vaccination efforts [190].

Globally, about 3 billion doses of FMD vaccines are used annually, mainly in China and South America [191]. These vaccines, primarily composed of binary ethyleneimine (BEI)-inactivated purified antigens depleted from viral NS proteins and combined with adjuvants, are available either as monovalent or multivalent formulations [192]. Despite their effectiveness in eradicating FMD in numerous regions and managing its spread in others, current vaccines are far from perfect. For instance, vaccine production requires expensive high-containment facilities due to the risk of viral escape, and the vaccine’s heat sensitivity demands a maintained cold chain to ensure viability from manufacture to administration [192,193]. Since there is little cross-protection between serotypes, matching between field and vaccine strains is critical for effective FMD control programs, and in some cases, new strain-specific vaccines must be developed by adapting field viruses to grow in cell culture [10]. While these vaccines are designed to prevent clinical disease, they do not completely inhibit viral replication in the nasopharynx, potentially leading to carrier animals that harbor the virus long-term [136,194]. Additionally, the DIVA characteristics of current vaccines rely on antigen purification strategies to minimize NS viral proteins [195]. Most importantly, a significant drawback is the seven-day period required for the vaccine to confer protective immunity [11,196,197], creating a vulnerability window in which rapid viral spread is possible, especially in FMD-free countries in which animals are fully susceptible to the virus.

Given these challenges, several groups throughout the world are dedicated to improving current vaccines and developing new platforms to address these gaps. In this section, we explore these innovative platforms, highlighting their benefits and constraints in the context of vaccine development (Figure 2).

### 7.1. Inactivated Vaccine Platform

The inactivated vaccine for FMD is a crucial and effective platform in mitigating the disease. However, efforts are ongoing to enhance the current inactivated vaccine platform. Some of these efforts focus on developing more permissive cell lines to increase the production yields of vaccine strains, thereby reducing the need for the serotype-specific adaptation of FMDV vaccine seedstocks. This has been shown in recent reports by LaRocco et al. [9,198] and Harvey et al. [10], which have demonstrated that the overexpression of αvβ6 integrin—a natural receptor for FMDV [129]—in cell lines can increase susceptibility to all FMDV serotypes. In parallel, there is a strong effort to improve capsid stability, which is critical for the effectiveness of vaccines since the strength of the immune response is proportional to the number of intact 146S particles in the vaccine [199]. One possibility is to introduce mutations in the capsid-coding region to produce virus particles that are more resistant to temperature fluctuations and changes in pH levels, with promising results recently reported [15,16,17,18,200]. Complementing these genetic strategies, advancements in vaccine formulation are also being evaluated. The use of stabilizers, such as trehalose, NaCl, and CuSO_4_·5H_2_O, has shown promise in prolonging vaccine shelf-life [201,202].

Predicting the correct vaccine matching has always been an important topic [203,204], but it can cause significant delays in outbreak response. To mitigate this, a high-potency vaccine with a homologous potency of 6 PD50 (50% protective dose) is recommended during outbreaks (Brehm KE et al., 2008 [11]). Recent studies have explored protection offered by higher-potency FMD commercial vaccines and their ability to prevent disease spread (Dekker et al., 2020 [12]; Galdo Novo et al Vaccine, 2018 [13]; Fishbourne et al., 2017 [14]). Understanding cross-protection mechanisms is crucial for evaluating the performance of FMD vaccination in the field [205]. Research by Li et al. [22] demonstrated that a genetically modified FMDV vaccine, developed using full-length cDNA clones and incorporating amino acid changes in the most antigenic site of the virus, exhibited enhanced efficacy against the various FMDV O topotypes prevalent in China. This was observed when the vaccine was evaluated in pigs after the vaccine was inactivated and formulated with oil. This finding underscores the vaccine’s potential for broad cross-protection.

Furthermore, research is increasingly focusing on cross-protection in both mono- and multivalent vaccine formulations, with a particular interest in not only intra-serotypic variations [12,206,207,208] but also the development of evaluation methods that aim to highlight the qualitative measures of vaccine protection, particularly looking at the strength and specificity of vaccine-induced antibodies [209,210,211]. The generation of chimeric FMDVs with antigenic properties that can be engineered to display epitopes from diverse FMDV variants represents a possible path toward enhancing cross-protection, as demonstrated with viruses circulating in South Africa and China [19,20,212]. A step further in this concept is the creation of polyvalent vaccine antigens composed of mosaic proteins. These proteins are assembled from fragments of multiple natural sequences using a computational optimization method to maximize the representation of potential epitopes across various viral populations. This strategy, which was originally applied to influenza vaccines [213], is now being explored for the prevalent FMDV serotypes A, O, and Asia1, potentially leading to a more robust and comprehensive protective response against FMDV (personal communication). Notably, delivering vaccination strategies to endemic regions like Africa remains a challenge, with inactivated vaccines falling short of reaching optimal levels to control the disease [214]. The efficacy of an inactivated vaccine is significantly enhanced by the use of safe and effective adjuvants. Traditionally, the FMD vaccine has been combined with mineral oil adjuvants like Montanide^®^ ISA 201 or 206 (Seppic Inc., Castres, France), prepared as water-in-oil-in-water emulsions or aluminum-based (alum/saponin) adjuvants [215]. However, new investigations in this field are reporting important advancements, such as the development of Montanide ESSAI IMS D 12802 VG PR, which has demonstrated the capacity to shorten the time required to establish protection [216]. Other explored adjuvants, including CAvant ^®^ SOE (CA V AC, Daejeon, Korea) [217], non-coding RNA [218], and a variety of natural Chinese medicine-based formulations [219,220], have recently been reported. For a more comprehensive overview of these developments, refer to reference [221].

Safety is paramount in the production of inactivated vaccines to prevent the escape of virulent strains from manufacturing facilities. This risk has led many countries to rely on imported vaccines rather than produce them domestically. Innovations in vaccine strain development—creating strains that are attenuated in animals yet viable in cell culture—may pave the way for safer, next-generation FMD-inactivated vaccines. Notably, a leaderless FMDV strain (see Section 2 above) was removed from the United States Select Agent Program regulations in April 2018 (Centers for Disease Control Prevention, 2023, January, Attenuated Strains of USDA-only Select Agents Excluded, Section 121.3. https://www.selectagents.gov/sat/exclusions/index.htm, accessed on 20 August 2024); this represents a major step forward because it reduces the requirements for policies and procedures designed to maintain security. These strains are currently under development in the United States, aiming to produce high-potency vaccines that are fully DIVA-compatible. This progress holds the promise of enhancing biosecurity and self-reliance in the FMD vaccine supply.

### 7.2. Virus-like Particle Vaccine Platform

Virus-like particles (VLPs) consist of viral structural proteins that self-assemble (empty capsids) within cells, resembling the live virus in appearance but lacking the capacity for replication. The VLP vaccine platform offers a clear advantage over the current vaccine by eliminating the need to grow infectious virus, thus reducing biosafety risks. VLPs, which preserve the virus structural immunogenicity needed to induce protective humoral immunity, are readily recognized by the immune system and can stimulate dendritic cells in the same way as inactivated FMDV [222]. There are generally two design approaches for VLP vaccines. Empty capsids can be produced within the vaccinated host from capsid genes carried by a viral vector (also called subunit vaccines) or in various culture systems, including bacteria, yeast, mammalian cells, insect cells, or plants, before being administered as a vaccine.

Efforts to develop subunit vaccines for FMDV have utilized a spectrum of viral vectors, including vaccinia virus [25], fowlpox virus [26], attenuated pseudorabies virus (PRV) [27] (219) or the single-cycle Semliki Forest virus (SFV) [28], and adenovirus (Ad5) [29], to induce protection against FMDV. The Ad5-FMD vaccine in particular is a licensed emergency-use vaccine in the US for FMD type A [32], and it has shown promise as an alternative to inactivated virus vaccines for worldwide use. This platform has demonstrated early protection in swine and cattle against different FMDV serotypes [30,223], with protection lasting for up to 42 days after vaccination [224]. The Ad5-FMD vaccine is distinguished by its safety profile, DIVA capabilities, and stability as well as the ability to eliminate the need for adapting field strains for vaccine production. However, the requirement for high doses to achieve protective responses makes production costly. Enhancements like adding the full-length 2B coding region to increase the synthesis of FMDV capsid proteins [31,94], inserting an extra RGD motif to the Ad5 fiber coding protein sequence for dendritic cell targeting [225], and formulating the vaccine with a synthetic double-stranded RNA stabilized with poly-L-lysine and carboxymethyl cellulose (polyICLC) [226] have proven to be successful strategies to enhance the potency of Ad5-FMD vaccines.

The most common bacteria expression system for the generation of FMDV VLPs is *Escherichia coli* [39,227], with other systems such as *Salmonella typhimurium* [34] and *Lactococcus lactis* [35] also reported. To enhance the immunogenicity and yield of these VLPs in swine, strategies such as amino acid substitutions have been employed to strengthen hydrophobic interactions within the capsid [221]. Additionally, yeasts, particularly *Hansenula polymorpha* [228] and *Saccharomyces cerevisiae* [229], have been utilized as an expression platform, with studies showing that FMDV VLPs produced in this yeast, when formulated with aluminum and CpG as adjuvants, are immunogenic [228]. Most recently, Song et al., 2024 [40] developed a VLP-based combined vaccine strategy targeting FMDV serotypes A and O as well as Seneca virus (SVA)—another picornavirus that affects livestock animals. These vaccines have shown the ability to induce strong neutralizing antibody responses in pigs, highlighting their broad-spectrum capabilities and cost-effective features.

Over the years, significant efforts have been made to produce VLPs in insect cells using the baculovirus expression system. Techniques to stabilize capsids and modify expression in this system have been refined, achieving protective responses in cattle [19,37,38]. Plant-based VLP expression is another innovative approach, as animals could be immunized by feeding them crops like alfalfa, tomato, or tobacco expressing the empty capsids. However, the effectiveness of this approach in the natural host remains to be demonstrated [41,42].

The expression of VLPs in mammalian cells has also become popular [230] for various reasons, including the ability to introduce accurate post-translational modifications, which are crucial for the structural integrity and function of proteins [231]. The ability to clone sequences from various FMDV serotypes into expression plasmids allows for rapid adaptation to newly emerging viral strains. This technology has been successfully applied to produce VLPs that confer protection in cattle and swine under experimental conditions [43], and the selection of the right adjuvant can have a great impact on the immunogenicity of the VLPs [232].

### 7.3. Synthetic Peptide and DNA Vaccine Platforms

The rationale behind peptide-based vaccines is based on the accurate prediction of T-cell and B-cell epitopes that are known to elicit specific and protective immunity. This approach enables the engineering and refinement of epitope structures—mainly derived from immunodominant sequences—to increase their potential for inducing robust immune responses [233]. Furthermore, synthetic peptides offer a non-infectious alternative to conventional vaccines, with high stability and purity levels that can exceed 95%, providing a significant advantage over traditional inactivated vaccines. Multi-epitope-based vaccines have achieved success in pigs in China [234]; however, they have not been used in ruminants due to the limited efficacy observed in these species [44,235]. While the synthetic FMD peptide vaccine is extensively utilized in swine agriculture throughout China, there is a need to revise the peptide formulation to confer protection against emergent viral lineages. Notably, peptides typically have low immunogenicity, presenting a challenge for development. Innovative strategies to address these challenges, such as the use of multiple antigenic peptides (MAPs) or dendrimers (complex branched macromolecules with a peptide core attached to multiple epitopes), have shown promise. These structures, particularly when combined with a water-in-oil single-emulsion adjuvant, have successfully conferred protection in swine [46] and, to a lesser extent, in cattle [47], establishing them as potential candidates for a subunit DIVA vaccine against FMDV. Enhancing protection in cattle could be possible with the use of an adjuvant containing nanoliposomes, as described by Heshmati et al. [236] in small animal models. Other approaches to improve the effectiveness of this vaccine platform include the combinatorial use of adjuvants, such as the one provided by polyinosinic and polycytidylic acid (poly (I:C)) [234]. Most recently, the use of recombinant T4 phage nanoparticles displaying FMDV epitopes has shown potential as a cost-effective and highly immunogenic vaccine without the need for external adjuvants, offering a promising alternative [237].

Likewise, the DNA vaccine platform offers a safer alternative to inactivated vaccines, with DIVA properties and the flexibility for the rapid incorporation of sequences from emerging field strains, potentially expressing multiple antigenic sites from different serotypes. Additionally, DNA vaccines can be produced at a lower manufacturing cost and allow for adaptable design and construction. However, challenges remain regarding DNA stability and the level of antigen expression; large amounts of DNA and multiple inoculations are needed to induce a relatively low FMDV-specific neutralizing antibody response with variable protection levels [49,238,239]. Recent improvements include the co-expression of FMDV B and T-cell epitopes with an antiapoptotic protein (i.e., Bcl-xL), which improved the T-cell response in a mouse model for FMD [50]. Additionally, a DNA vaccine based on mannosylated chitosan nanoparticles (which are known to enhance intracellular transport) or a self-replicating gene vaccine was evaluated in guinea pigs, demonstrating substantial improvement in all the immunological parameters with enhanced protection [240,241]. Altogether, the applicability of these approaches will remain limited until studies with the natural host are performed.

### 7.4. Modified Live-Attenuated Vaccine Platform

Modified live-attenuated vaccines (MLAVs) have played an important role in the eradication or near-eradication of various diseases in humans and animals [242,243]. These vaccines effectively mimic the natural immune response to viral infection, leading to rapid and long-lasting protection. Genetic engineering has enhanced our ability to generate attenuated strains by either deleting important regions of the viral genome or introducing mutations into known virulence genes. As previously noted (see the virulence factors section), various attenuated FMDV strains carrying deletions or modifications in Lpro have shown success as potential vaccine candidates.

Other efforts have involved the substitution of FMDV Lpro with that from another *Aphtovirus*, such as bovine rhinitis B virus (BRBV). This substitution resulted in the generation of a chimeric FMDV that showed an attenuated phenotype in cattle and induced protective immunity against challenge with wild-type homologous FMDV [23,244]. However, like the case of the FMDV SAP mutant (see virulence factor section), the possibility of reversion by recombination with circulating wild-type FMDV makes it a non-ideal candidate.

To enhance the safety profile of FMDV live-attenuated vaccine candidates and minimize the risk of reversion, codon deoptimization technology has been utilized to design attenuated strains. The recoding of the viral genome through codon deoptimization has been shown to increase CpG and UpA frequencies, which can enhance the engagement of innate immune responses and attenuate viral infections [245,246,247]. Initial research showed that the deoptimization of the P1 coding region resulted in severe attenuation in vitro and in vivo, with the induction of high titers of neutralizing antibodies in swine [58]. Although this approach led to a marked attenuation across various FMDV serotypes, it did not consistently induce a sufficiently robust adaptive immune response to confer protection [60]. A deeper understanding of the molecular determinants of attenuation when recoding the FMDV genome will aid in calibrating the optimal degree of attenuation. On the other hand, targeting more conserved regions of the FMDV genome for deoptimization could allow for the creation of a stable viral backbone. Such a backbone could be universally applied to generate chimeras with the P1 structural regions of different serotypes. Recent work by Diaz-San Segundo et al. [59] showed that the deoptimization of the non-structural P2/P3 regions is well-tolerated and triggers an adaptive immune response in both mice and swine. Interestingly, this approach, which includes DIVA markers, yielded a highly attenuated virus in cattle that also induced a strong protective immune response (manuscript in preparation), presenting an attractive platform for vaccine development. Nevertheless, additional research is required to evaluate the longevity of the immune response and to examine other safety parameters, such as the risk of recombination, before this strategy can be advanced as a commercial vaccine candidate.

### 7.5. Potential Use of Novel Technologies

One of the most novel vaccine platforms recently developed is based on the use of messenger RNA (mRNA), which is quickly becoming a highly effective technology for producing vaccines and biomedical therapeutics for emerging infectious diseases. This is largely due to its short manufacturing lead time and economic scale-up production [248]. Unlike traditional vaccines, mRNA-based vaccines do not require an infectious virus or cell culture, overcoming several limitations [249]. To date, all mRNA vaccines against animal pathogens have been explored for the prevention of zoonoses, like rabies [250], tick-borne encephalitis [251], and Zika [252]. In the case of FMDV, early research showed that immunization with the full-length, genetically engineered FMDV mRNA could induce strong immune responses to FMDV in a mouse model, suggesting the potential for FMDV in natural hosts [253]. However, using the full-length viral genome presents the risk of infectivity in natural hosts. Since there is evidence that the empty capsid vaccine can be generated by the co-expression of the FMDV P1-2A capsid precursor plus the 3Cpro via baculovirus expression systems to effectively protect animals against FMDV [254], it is possible to speculate that a synthetic, thermostable modified RNA vaccine encoding the FMDV P1-2A and 3Cpro genes could be engineered to protect livestock from emerging FMD strains.

A promising technology based on the use of plasmid DNA-based FMDV minigenome was recently described [255]. When these plasmids are transfected into cells, they efficiently produce uncapped mRNAs in the cytoplasm, which then initiate the synthesis of viral genomic RNA-like molecules, leading to the creation of infectious FMDV. By applying a strategy similar to that used for mRNA vaccines, these minigenomes, which express only the FMDV P1-2A and 3Cpro genes, could be utilized to generate immunogenic FMDV VLPs tailored to distinct serotypes. However, further studies are needed to validate this approach.

## 8. Conclusions and Future Directions

The search for effective FMDV vaccines is significantly challenged by the virus serotype diversity, including six circulating serotypes and numerous subtypes, each demanding tailored vaccines for each variant, as cross-protection is limited. This complexity is worsened by the virus’s broad host range and high mutation rates, which lead to the continuous generation of new variants that may resist the immunity provided by existing vaccines. Moreover, traditional vaccination protocols require frequent booster doses to sustain immunity, adding to the logistical and economic burdens. An additional concern is the potential for vaccinated cattle to become carriers, perpetuating virus transmission. The requirement of a cold chain to preserve most FMDV vaccines introduces further complications in terms of transport and storage, particularly in regions where FMD is endemic. Furthermore, existing vaccines do not align with DIVA capabilities, creating a significant problem for epidemiological surveillance and control efforts. To address these issues, efforts are underway to develop novel vaccine platforms, many of which show promising results in controlling FMDV. Beyond vaccination, effective disease control also requires robust surveillance programs, comprehensive economic analyses from governments, and the overcoming of educational and cultural barriers to enhance FMD management on a global scale. While these aspects are beyond the scope of this review, they are important to acknowledge

As we look to the future, it is imperative to develop next-generation FMDV vaccines that can overcome these hurdles. In the realm of vaccine innovation, LAVs have emerged as a promising approach, as they have historically paved the way for the successful control and eradication of various viruses, including yellow fever, Rinderpest, and several human respiratory viruses, such as measles and rubella [242,243,256,257]. These vaccines are known to elicit broader and long-lasting protective humoral and cell-mediated immunity due to their ability to replicate, which can engage the host mucosal along with systemic immune defenses. However, using this strategy for FMDV entails complexities surrounding safety and the inherent genetic stability of the virus, which must be addressed to ensure viability. Notably, in recent years, LAV candidates for FMDV with DIVA capabilities have been developed using codon deoptimization strategies, offering promising outcomes in experimental studies [58,59,60]. Ongoing studies are critical to evaluate the safety profile in multiple animal species and the broader applicability across multiple FMDV-susceptible species, refine strategies to achieve cross-protection against diverse serotypes, and evaluate the vaccine capacity to stimulate mucosal immunity. Before considering a particular LAV candidate, it is also essential to conduct studies on transmission from animals inoculated with LAV candidates to naive in-contact animals of the same species including wildlife species and under different experimental conditions reflecting livestock production environments. Additionally, the potential issue of recombination events between attenuated FMDV strains and wild-type strains must be addressed through in vivo studies. Government perspectives on implementing such a strategy may be cautious unless comprehensive safety studies are carried out.

A clear approach to overcoming safety concerns with vaccines is the use of subunits or VLPs, which have demonstrated encouraging outcomes. Advancements in the understanding of FMDV biology have enabled the identification of amino acids within the virus’s structural proteins that can be manipulated to increase thermostability, which can solve problems of storage and transport in countries where the disease is endemic. However, the use of these subunit- or VLP-based platforms often requires higher doses and multiple boost immunizations or supplementation with adjuvants to achieve the desired level of immunogenicity. Similarly, multi-epitope vaccines offer a dual advantage of safety and the potential to open the door toward the world of a “universal” vaccine. These vaccines can be tailored to provide protection against a wide range of serotypes and subtypes in a single dose. The concerted efforts of numerous research groups are dedicated to enhancing and refining this innovative platform [258,259].

Enhancing our understanding of the diverse immune compartments and the various immune cells that interact with FMDV or are stimulated by it during infections in swine and cattle, beginning at mucosal surfaces and subsequently spreading systemically, is essential for the informed development of vaccines or adjuvants. To this end, employing multi-omic strategies, such as transcriptomic and proteomic analyses, can deepen our insight into virus–host interactions. These approaches should provide valuable information about the changes in gene expression and protein profiles experienced by immune cells during infection or in response to vaccination.

## Figures and Tables

**Figure 1 vaccines-12-01071-f001:**
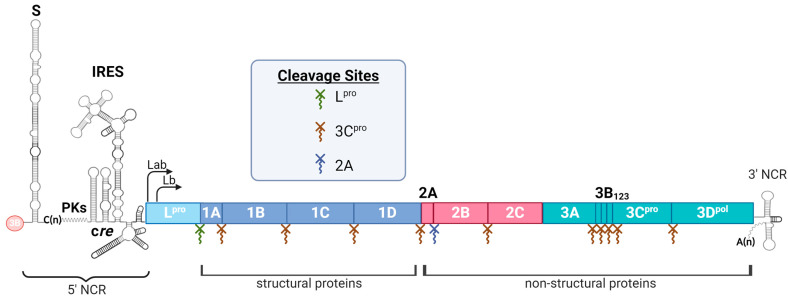
FMDV Genome organization. RNA elements are shown at the 5′ and 3′ non-coding regions (NCRs). The 5′ NCR contains several segments with secondary RNA structure, including S fragment, poly (C) tract (C(n)), pseudoknots (PKs), cis-acting replicative element (cre), and internal ribosome entry site (IRES). The 3′ NCR consists of a short stretch of RNA with two elements of predicted secondary structure and a poly (A) sequence (A(n)). The 3B (VPg) protein is shown covalently linked to the 5′ end of the genomic RNA. The ORF is depicted as an outlined box filled with multiple colors. The processing sites for 3C^pro^, 2A, and L^pro^ are indicated by brown, blue, and green symbols, respectively. Arrows describe the two AUG initiation codons, Lab and Lb. Illustration was created with Biorender.com.

**Figure 2 vaccines-12-01071-f002:**
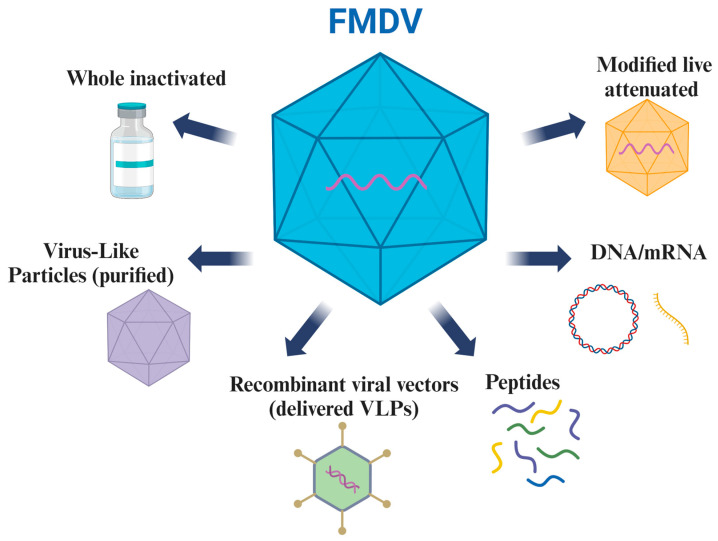
Current vaccine platforms under investigation to protect against FMD. Created with Biorender.com.

**Table 1 vaccines-12-01071-t001:** Foot-and-mouth vaccine platforms under investigation and development.

	Type	Strategies for Vaccine Improvement	References of Interest	Coding Region Targeted
Inactivated vaccines	Current inactivated vaccines	More permissive cell lines for viral growth	LaRocco et al., 2021 [9]; Harvey et al., 2022 [10]	n/a
		High Potency	Brehm KE et al., 2008 [11], Dekker et al., 2020 [12], S Galdo Novo [13], Fishbourne et al., 2017 [14]	n/a
		Capsid stabilization	Scott et al., 2017 [15]; Dong et al., 2021 [16]; Lopez-Arguello et al., 2019 [17]; Yuan et al., 2020 [18]	P1 capsid coding region
		Enhancing cross-protection by reverse genetic, chimeric, and mosaic capsid designs	Kotecha et al., 2018 [19]; Li et al., 2022 [20]; Rieder et al., personal communication [21], Li et al., 2012 [22]	P1 capsid coding region
	New marked inactivated vaccines	Development of avirulent FMDV strains with DIVA (differentiation between infected and vaccinated animals) markers in different NS proteins (Lpro, 3AB) that are safer for production	Uddowla et al., 2012 [23]; Hardham et al., 2020 [24]	Lpro, 3B, and 3D
Virus-like particles expressed by viral vectors		Improve safety and include DIVA capabilities, possibly decreasing costs using:		
	Vaccinia	Vaccinia virus to deliver FMDV empty capsids with controlled 3C expression	Steigerwald et al., 2020 [25]	P1 coding region, 3Cpro
	Avian poxvirus	Fowlpox virus expressing VLPs and swine IL-18	Ma et al., 2008 [26]	P1 coding region, 2A, 3Cpro
	Pseudorabies	Pseudo rabies V vector expressing VLPs	Hong et al., 2007 [27]	P1 coding region, 2A, 3Cpro
	Alphavirus	Single-cycle self-replicating RNA Semliki Forest virus vector expressing VLPs	Gullberg et al., 2016 [28]	P1 coding region, 2A, 3Cpro
	Adenovirus	Replication-defective human adenovirus type 5 (Ad5) expressing VLPs	Mayr et al., 1999 [29]; Moraes et al., 2002 [30]; Pena et al., 2008 [31]; Grubman et al., 2012 [32]; Schutta et al., 2016 [33]	P1 coding region, 2A, 2B, 3Cpro
	Bacterial vectors	*Salmonella typhimurium* or *Lactococus lactis* expressing VLPs	Zhi et al., 2021 [34]; Liu et al., 2020 [35]	VP1
Purified virus-like particles		More safety, DIVA capabilities, lower costs, and rapid adaptability to circulating strains:		
	Baculovirus	Purified VLPs expressed from recombinant baculovirus	Kotecha et al., 2015 [36]; Porta et al., 2013 [37]; Ganji et al., 2018 [38]	P1 coding region, 2A, 3Cpro
	Bacterial	Purified VLPs expressed in *E. coli*, VLP-based combined vaccine	Xiao et al., 2016 [39]; Song et al., 2024 [40]	VP0, VP1, VP3
	Plant	VLPs expressed in transgenic alfalfa, tomato fruits, or tobacco	Dus Santos and Wigdorovitz, 2005 [41]; Veerapen et al., 2017 [42]	P1 coding region, 2A,
	Mammalian cells	VLPs expressed in mammalian cell cultures	Puckette et al., 2022 [43]	P1 coding region, 2A, 3Cpro
Peptide vaccines	T- and B-cell peptide epitopes	Improve safety, include DIVA capabilities, and extend shelf life by producing:		
		Peptides combining VP1 G-H loop epitopes with T-cell epitopes	Wang et al., 2002 [44]; Rodriguez et al., 2003 [45]	VP1-GH-loop
		Dendrimeric peptides containing one T-cell epitope and four B-cell epitopes	Blanco et al., 2016 [46]; Soria et al., 2017 [47]	P1
DNA vaccines		Improve safety and include DIVA capabilities and rapid adaptability to circulating strains by:		
	Electroporation	Administering DNA by electroporation	Fowler et al., 2012 [48]	P1, 2A, 3C, 3D
	APC targeting	Using B/T cell epitopes fused to a single chain antibody or Bcl-xL anti-apoptotic signal	Borrego et al., 2011 [49]; Gülçe İz et al., 2013 [50]	P1
Modified live-attenuated vaccines		Improve safety and immune response (more rapid and sustained) by:		
	Mutations on Lpro	Introducing attenuating mutations in L^pro^ coding sequence	Diaz-San Segundo et al., 2012 [51]; Medina et al., 2020 [52]; Azzinaro et al., 2022 [53]	Lpro
	Chimeric virus	Substituting FMDV coding regions with other virus proteins (chimeric FMDV/bovine rhinitis B virus)	Uddowla et al., 2013 [54]	Lpro
	Fidelity	Introducing point mutations in 3Dpol to alter replication fidelity	Rai et al., 2017 [55]	3Dpol
	Untranslated region	Introducing mutilations targeting RNA structures in the non-translated region	Rodriguez-Pulido et al., 2009 [56]; Kloc et al., 2017 [57]	5′ or 3′ UTR
	Deoptimized virus	Deoptimization of P1 or P2/P3 coding regions	Diaz-San Segundo et al., 2015 [58]; 2020 [59]; Medina et al., 2023 [60]	Genome-wide

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
