# Peer review of "Virulence and Immune Evasion Strategies of FMDV: Implications for Vaccine Design"

_vaccines, 2024, doi:10.3390/vaccines12091071_

Round 1

Reviewer 1 Report (Previous Reviewer 1)

Comments and Suggestions for Authors

I notice that all of my review comments have been satisfactorily answered, and I have no further comments.  

Author Response

We appreciate the reviewer's comments.

Reviewer 2 Report (Previous Reviewer 4)

Comments and Suggestions for Authors

In “Inactivated Vaccines” section, inactivated vaccine is the most important preventive measure against foot-and-mouth disease. At present, the inactivated foot-and-mouth disease vaccine produced by Chinese scholars using reverse genetic technology can effectively inhibit FMDV replication and be exported to other countries. In addition, Chinese scholars have also produced VLPs vaccines and obtained new veterinary drug certificates. None of these information are included in the review. In the present review, the authors only summarized several vaccines at a large level, and there are a lack of effective and available information.

Author Response

We sincerely apologize for the oversight. We have now included a sentence that highlights the importance of the inactivated vaccine platforms and reads: "The inactivated vaccine for FMD is a crucial and effective platform in mitigating the disease. However, efforts are ongoing to enhance the current inactivated vaccine platform." We also have noted the specific research by Li et al., which demonstrates the use of reverse genetics and the incorporation of certain mutations with circulating FMDV strains in China, offering a pathway for the inactivated vaccine to provide broad cross-protection.  The paragraph now reads: “Research by Li et al. [170] demonstrated that a genetically modified FMDV vaccine, developed using full-length cDNA clones and incorporating amino acid changes in the most antigenic site of the virus, exhibited enhanced efficacy against the various FMDV O topotypes prevalent in China. This was observed when the vaccine was evaluated in pigs after vaccine was inactivated and formulated with oil. This finding underscores the vaccine potential for broad cross-protection."

In this review, we do not address the exportation of vaccines and will therefore refrain from discussing this topic. Additionally, we have incorporated the work of Guo et al., which highlights the use of E. coli for the purification of VLPs, as well as the latest report on the development of VLP-based combination vaccine strategies from Chinese scholars. The paragraph now reads: “Most recently, research by Song et al 2024 have developed “VLP-based combine vaccine” strategy targeting FMDV serotypes A and O, as well as Seneca virus (SVA) -another picornavirus that affects livestock animals. These vaccines have shown the ability to induce strong neutralizing antibody responses in pigs, highlighting their broad-spectrum capabilities and cost-effectiveness.”

Round 2

Reviewer 2 Report (Previous Reviewer 4)

Comments and Suggestions for Authors

The author addresses my comments in part.

This manuscript is a resubmission of an earlier submission. The following is a list of the peer review reports and author responses from that submission.

Round 1

Reviewer 1 Report

Comments and Suggestions for Authors

The general scientific description of FMDV in lines 10 to 15 should not be appeared in the “Abstract” of the article, and can be moved to the "Introduction" section, or deleted. Start with "In this review," but go further.

Although there are many difficulties to be overcome in the design of FMDV composite multi-epitope vaccine and universal vaccine, if successful, it will bring great convenience to FMDV prevention and control. Hope the authors will pay more attention to this aspect in this review.

Author Response

We thank the reviewer for the comments. In response, we have removed the general description of the virus (FMDV) while retaining the description of the disease (FMD) as we believe is beneficial for readers to have some background in the abstract to set the focus of the review. As per the reviewer’s suggestion, we have also shortened the initial sentences of the abstract.

We agree with the reviewer regarding the challenges in designing a composite multiepitope universal vaccine for FMDV. Throughout the manuscript, we have highlighted the difficulties posed by the diversity of virus serotypes and the need for tailored vaccines for each variant due to limited cross-protection (516-517; 824-825) and the different efforts in developing multi-epitope vaccine platforms (lines 634-635). To further address the reviewer’s suggestion, we have emphasized the importance of exploring innovative strategies to overcome these challenges and enhance the effectiveness of FMDV prevention and control efforts in the “Challenges and Future Directions Sections.  We have added the following paragraph: “Similarly, multi-epitope vaccines offer a dual advantage of safety and the potential to open the door toward the world of a “universal” vaccine. These vaccines can be tailored to provide protection against a wide range of serotypes and subtypes in a single dose. The concerted efforts of numerous research groups are dedicated to enhancing and refining this innovative platform (Shao et al., 2024; Chathuranga et al., 2022)"

Reviewer 2 Report

Comments and Suggestions for Authors

The work by Medina et al., describes the virulence and immune evasion of the virus FMDV and the strategies to design a vaccine against it. It is a very long review manuscript well-written and very informative. The chapters are well-structured with abundant references. The strategies to design vaccines are very promising and provide a framework for future research in the subject area. This review summarizes very clearly the last research on the area. I do not find any general or specific details to be pointed out. My recommendation is that it should be published. 

Author Response

We thank the positive comments of the reviewer. In response, we have condensed the manuscript while maintaining its overall structure and focus. 

Reviewer 3 Report

Comments and Suggestions for Authors

The paper by Gisselle Medina and Fayna Diaz San Segundo describing the Virulence and immune evasion strategies of FMDV: implications for vaccine design and countermeasure strategies, is well-written in terms of the English language, however it reads as a chapter of a thesis. The manuscript tries to cover every aspect from virus replication, infection of its host, immune response and vaccines, aspects that have been reviewed multiple time before. I believe the manuscript will have more impact if it displays a proper focus on a specific topic, for example the immune response in response to FMDV infection, the virus evasion strategies and how the information can be used to design better vaccines. This is the title of the manuscript and the information reflecting within the manuscript should support that topic.

The authors used 323 references. It is impossible for any reader to even remotely try to cross-check all these references for validity. However, it became clear from the onset of the manuscript the authors also struggle to correctly use references in support to statements. E.g. Drs Marvin Grubman and Barry Baxt in their 2004 review paper did not describe the possible extinction of serotype C (line 36, paragraph 1, Introduction). In 2021 the reference laboratories under leadership of Dr David Paton provided the first evidence of the possible extinction of serotype C. Also, it is unclear how the genome sequence paper of Carillo et al., 2005 support the statement of lack of cross-immunity or cross-protection stated in line 37, par.1, Introduction, since the information relates to a clinical study. The word "negligible" in line 37 may be misleading as studies by Bryan Charleston has shown that exposure of animals to different serotypes may elicit antibodies that do cross-react to serotypes it was not exposed too.

It is the responsibility of the authors to check all references to make sure it supports the statement in the manuscript it was referenced for. 

The reviewer strongly differs from the statement in lines 38-40 of the Introduction. FMDV in endemic regions is not causing underdevelopment in these countries. Significant amounts of government budgets are spent in human healthcare, war, protection of borders, crime, education, there is little money left for disease control of veterinary importance. Together with many other diseases that kills farm animals is what leads to economic losses and poverty. The statement should be corrected or removed.

The statement in line 42 of the Introduction, it is critical to understand that improved FMDV vaccine platforms will not lead to eradication of FMDV in endemic regions. FMDV is maintained in wildlife. The conservation of wildlife is clear about not interfering with natural processes, including disease cycle in wildlife. Even though all FMDV is eradicated from cattle and no cattle-to-cattle transmission cycle exist, there will always be transmission events from wildlife to cattle. The disease can be managed by combination of movement control, sanitary measures and vaccines, but not vaccines on its own. This statement is misleading and should be corrected.

Line 47, paragraph 2, Introduction. I assume the authors meant "....and teats."

Lines 48-50, Introduction. The authors referenced the transmission routes in the first half of the sentence. However, how these routes contribute to its virulence in the 2nd half of the sentence should also be referenced.

The sentence in lines 51-52 of the Introduction starting with "In tissue culture systems...." is out of place and do not support or add value to the infection dynamics in host animals. Also, the reference is wrong. Rather remove the sentence.

Line 56. The sentence read virus viral, e.g. "FMDV viral virulence..." Rather state "FMD viral virulence."

Line 58. "various phylogenetically related FMDV variants (quasispecies)" should be changed to "various genetically related FMDV variants (quasispecies)" as the definition of quasispecies described the population structure based on genome variants related by mutations. 

Page 5, par.1, lines 73-75. The statement ending with "... in blocking immune responses at multiple levels." need to be referenced. 

Page 11, lines 426-427. It is not convincing that any of the references provided support the statement that macrophages are infected by FMDV, however, FMDV-Ab complexes are rather internalized by macrophages via antibody-mediated phagocytosis.

Page 13, section 4, lines 551-552. What about the need for vaccines in the remainder of endemic region. Africa for example require roughly 3 billion doses of vaccine to control the disease, while less than 5% of the need are actually applied.

Page 20, section 6. It is unclear from this literature study how hurdles, potential risk and governments viewpoints on the safety of LAV for FMDV will be overcome. It will add value to the manuscript if attention to the latter is provided.

The authors may consider to split the paper into 2 or 3 topics/papers. It could be a much better value-added effort and easier for non-FMD expert readers. The summaries of the immune response and vaccine strategies for FMDV control is thorough and should be published in one form or the other.

Kind regards

Comments on the Quality of English Language

No additional comments.

Author Response

Response to Reviewer 3 Comments

Summary

The paper by Gisselle Medina and Fayna Diaz San Segundo describing the Virulence and immune evasion strategies of FMDV: implications for vaccine design and countermeasure strategies, is well-written in terms of the English language, however it reads as a chapter of a thesis. The manuscript tries to cover every aspect from virus replication, infection of its host, immune response and vaccines, aspects that have been reviewed multiple time before. I believe the manuscript will have more impact if it displays a proper focus on a specific topic, for example the immune response in response to FMDV infection, the virus evasion strategies and how the information can be used to design better vaccines. This is the title of the manuscript and the information reflecting within the manuscript should support that topic.

We appreciate the reviewer’s insightful suggestion. We agree that maintaining a focused approach is crucial, and we will now highlight the virulence and immune evasion strategies to enhance the design of vaccines. In response, we have removed the section on countermeasures in the form of biotherapeutics, and have concentrated solely on vaccine methodologies against FMD. We have also modified the title slightly to remain focused. Furthermore, we have made additional revisions to ensure that our review remains centered on these viral evasion strategies.

The authors used 323 references. It is impossible for any reader to even remotely try to cross-check all these references for validity. However, it became clear from the onset of the manuscript the authors also struggle to correctly use references in support to statements. E.g. Drs Marvin Grubman and Barry Baxt in their 2004 review paper did not describe the possible extinction of serotype C (line 36, paragraph 1, Introduction). In 2021 the reference laboratories under leadership of Dr David Paton provided the first evidence of the possible extinction of serotype C. Also, it is unclear how the genome sequence paper of Carillo et al., 2005 support the statement of lack of cross-immunity or cross-protection stated in line 37, par.1, Introduction, since the information relates to a clinical study. The word "negligible" in line 37 may be misleading as studies by Bryan Charleston has shown that exposure of animals to different serotypes may elicit antibodies that do cross-react to serotypes it was not exposed too.

We thank the reviewer for looking at the manuscript so meticulously. We apologize for the oversight, we have revised and cross-referenced all the citations to properly indicate the references. We have added the reference for Paton et al 2021 and indicated the work of Yoon et al to support the statement of “lack of cross-immunity”. In addition, we have revised the sentence that contains the word “negligible” and now the sentence reads: “While clinical symptoms are roughly similar among subtypes, cross-immunity and cross-protection is limited, potentially contributing to recurrent outbreaks in endemic regions. Interestingly, studies such as the one by Grant et al provide insights into overcoming the challenges of cross-protection. Their research demonstrates that implementing a sequential vaccination approach with various FMDV serotypes can stimulate the production of antibodies that exhibit cross-reactivity with serotypes not directly encountered before the challenge.

It is the responsibility of the authors to check all references to make sure it supports the statement in the manuscript it was referenced for. 

We apologize for this oversight. We have now cross-referenced every single reference for validity.

The reviewer strongly differs from the statement in lines 38-40 of the Introduction. FMDV in endemic regions is not causing underdevelopment in these countries. Significant amounts of government budgets are spent in human healthcare, war, protection of borders, crime, education, there is little money left for disease control of veterinary importance. Together with many other diseases that kills farm animals is what leads to economic losses and poverty. The statement should be corrected or removed.

We agree with the reviewer’s comment and have remove the statement accordingly.

The statement in line 42 of the Introduction, it is critical to understand that improved FMDV vaccine platforms will not lead to eradication of FMDV in endemic regions. FMDV is maintained in wildlife. The conservation of wildlife is clear about not interfering with natural processes, including disease cycle in wildlife. Even though all FMDV is eradicated from cattle and no cattle-to-cattle transmission cycle exist, there will always be transmission events from wildlife to cattle. The disease can be managed by combination of movement control, sanitary measures and vaccines, but not vaccines on its own. This statement is misleading and should be corrected.

We thank the reviewer for the suggestion. We have now changed the word “eventual” to “potential” to note that it may not be possible with vaccines alone. In addition, we have added the following statement: “However, it is crucial to highlight that FMDV maintains a natural reservoir in the wild, specifically in the African buffalo, thereby increasing the likelihood of transmission events between wildlife and cattle. To effectively address this issue, it is imperative to implement additional mitigation strategies such as regulating movement, enhancing sanitary measures, and administering vaccines”

Line 47, paragraph 2, Introduction. I assume the authors meant "....and teats."

Thank you for pointing this out. We have addressed the typo.

Lines 48-50, Introduction. The authors referenced the transmission routes in the first half of the sentence. However, how these routes contribute to its virulence in the 2nd half of the sentence should also be referenced.

Thank you for the suggestion. We have now included a reference.

The sentence in lines 51-52 of the Introduction starting with "In tissue culture systems...." is out of place and do not support or add value to the infection dynamics in host animals. Also, the reference is wrong. Rather remove the sentence.

Thank you for pointing this out. We have removed the sentence.

Line 56. The sentence read virus viral, e.g. "FMDV viral virulence..." Rather state "FMD viral virulence."

Thank you for the suggestion, we have made the change accordingly.

Line 58. "various phylogenetically related FMDV variants (quasispecies)" should be changed to "various genetically related FMDV variants (quasispecies)" as the definition of quasispecies described the population structure based on genome variants related by mutations. 

Thank you for the suggestion. We have corrected the sentence accordingly.

Page 5, par.1, lines 73-75. The statement ending with "... in blocking immune responses at multiple levels." need to be referenced. 

Thank you for the suggestion. We have incorporated a few references. Gao, Y.; Sun, S.Q.; Guo, H.C. Biological function of Foot-and-mouth disease virus non-structural proteins and non-coding elements. Virol J 2016, 13, 107, doi:10.1186/s12985-016-0561-z. Rodriguez Pulido, M.; Saiz, M. Molecular Mechanisms of Foot-and-Mouth Disease Virus Targeting the Host Antiviral Response. Front Cell Infect Microbiol 2017, 7, 252, doi:10.3389/fcimb.2017.00252.

Page 11, lines 426-427. It is not convincing that any of the references provided support the statement that macrophages are infected by FMDV, however, FMDV-Ab complexes are rather internalized by macrophages via antibody-mediated phagocytosis.

Thank you for the suggestion. In response the sentence now reads: “Additionally, macrophages (Mφ) [111] and dendritic cells (DCs)  can internalized FMDV via antibody-complexes [112]. Understanding how FMDV affects these innate immune cells is critical for enhancing vaccine designs by improving: early immune responses, innate to adaptive immunological transitions, antigen presentation, and cytokine environment.

Page 13, section 4, lines 551-552. What about the need for vaccines in the remainder of endemic region. Africa for example require roughly 3 billion doses of vaccine to control the disease, while less than 5% of the need are actually applied.

Thank you for pointing this out. In response we have added a sentence: “Notably, delivering vaccination strategies to endemic regions like Africa remains a challenge, with inactivated vaccines falling short of reaching optimal levels to control the disease [181].”

Page 20, section 6. It is unclear from this literature study how hurdles, potential risk and governments viewpoints on the safety of LAV for FMDV will be overcome. It will add value to the manuscript if attention to the latter is provided.

Thank you for the suggestion. We have added a few sentences indicating this aspect: “Notably, in recent years LAV candidates for FMDV with DIVA capabilities have been developed using codon deoptimization strategies, offering promising outcomes. Ongoing studies are critical to evaluate the safety profile in multiple animal species, the broader applicability across multiple FMDV susceptible species, refine strategies to achieve cross-protection against diverse serotypes, and evaluate the vaccine capacity to stimulate mucosal immunity. Before considering a particular LAV candidate, it is also essential to conduct transmission studies from animals inoculated with LAV candidates to naive in-contact animals of the same species including wildlife species and under different experimental conditions reflecting livestock production environments. Additionally, the potential issue of recombination events between attenuated FMDV strains and wild-type strains must be addressed through in vivo studies. Government perspectives on implementing such a strategy may be cautious unless comprehensive safety studies are carried out.

The authors may consider to split the paper into 2 or 3 topics/papers. It could be a much better value-added effort and easier for non-FMD expert readers. The summaries of the immune response and vaccine strategies for FMDV control is thorough and should be published in one form or the other.

Kind regards

We thank the reviewer for this comment. In response, we have removed the section on countermeasures in the form of biotherapeutics, and have concentrated solely on vaccine methodologies against FMD

Reviewer 4 Report

Comments and Suggestions for Authors

Gisselle N. Medina et al summarized the virulence and immune evasion strategies of FMDV. Indeed, foot-and-mouth disease (FMD) is one of the most economically devastating diseases in the livestock industry worldwide. The relevant research of FMD is helpful to improve the prevention and control measures of FMD.

However, this review is missing a great deal of information about innate immunity and inactivated vaccines. For example, the 2B protein can reduce the immune response mediated by multiple host proteins, not just RIG-I and MDA5; inactivated vaccines for FMD are now recognized as the most important preventive measure against foot-and-mouth disease, but the authors have summarized only a small amount of information about inactivated vaccines in this review. These summaries by the author may mislead the reader.

In addition, the review only has one table and one figure, which makes it difficult for readers to understand the meaning of the article.

Overall, this is a low-quality review.

Author Response

Gisselle N. Medina et al summarized the virulence and immune evasion strategies of FMDV. Indeed, foot-and-mouth disease (FMD) is one of the most economically devastating diseases in the livestock industry worldwide. The relevant research of FMD is helpful to improve the prevention and control measures of FMD.

However, this review is missing a great deal of information about innate immunity and inactivated vaccines. For example, the 2B protein can reduce the immune response mediated by multiple host proteins, not just RIG-I and MDA5; inactivated vaccines for FMD are now recognized as the most important preventive measure against foot-and-mouth disease, but the authors have summarized only a small amount of information about inactivated vaccines in this review. These summaries by the author may mislead the reader.

 In addition, the review only has one table and one figure, which makes it difficult for readers to understand the meaning of the article. 

Overall, this is a low-quality review.

We are sorry to disappoint this reviewer as we have made efforts to include as much as possible of the literature available. Following the reviewer’s suggestions, we have made modifications throughout the manuscript to improve its quality. For instance, we have included a new figure of the viral genome, significant for the reader to understand the importance of the different proteins and non-coding regions addressed in the manuscript. We have made additional revisions to ensure that our review remains centered on viral evasion strategies and the implications in vaccine design. Also, we have revised the manuscript and expanded on the FMDV 2B as suggested by the reviewer. However, with respect to inactivated vaccines, we not only have explained the importance of vaccination campaigns using currently approved inactivated vaccine but have also dedicated a whole section to compile how multiple groups around the world are trying to improve this vaccine platform, so we are hoping that we do not mislead the reader.

Reviewer 5 Report

Comments and Suggestions for Authors

vaccines-2948251

 Virulence and immune evasion strategies of FMDV: implications for vaccine design and countermeasure strategies

By Gisselle Nilda Medina *, Fayna C Diaz San Segundo

The review discusses the diversity in virulence and immune responses against foot-and-mouth disease virus (FMDV) in susceptible host species, to serve as a foundation for a rational design of improved vaccines and countermeasure strategies, ideally with the aim to eradicate FMD.

The review is well written and informative. As a minor flaw, it tends to take for granted that the reader has at least basic knowledge of details of FMDV. In addition, it lacks figures. A figure of the NCR, or even of the whole genome, with its salient portions highlighted would be useful. I had to look one up online to help follow.

Major

Lines 88-89. a brief explanation of what the S segment is would be helpful.

Line 163-165. Were the pigs infected at all? Why is this interesting? Was the virus not infectious any more?

Lines 214-219 should be at the beginning of the paragraph to introduce protein 2c.

Line 277 (and Table 1) please explain what DIVA markers are, at least the acronym.

MINOR Comments

Table 1. I would suggest to add a column just for references rather than placing them after the improvement strategy. The journal style would be a number.

Line 94 attenuating should be attenuated

Line 101 Lpro should be introduced here

Line 135 impact on….

Line 151 “the infectious clone“ name is missing.

Line 159 effective should be introduced effectively.

Line 160 a virus should be a viral mutant.

Line 171 vaccine efficacy should be introduced efficacy as vaccines.

Line 172 this virus: which?

Line 320 In fact should be indeed.

Line 349 high  should be highly.

Line 368 should species not be serotype? Or are the authors referring to animal species?

Line 391 similar should be similarly.

Line 443 …known for their potent APCs should be …known for being potent APCs.

Line 456 Although, as in other DCs populations, FMDV infection of LCs is abortive. Meaning is not clear

Line 623 ..was removed… do the Authors imply that therefore its use is not restricted? Pls explain why this is an advantage.

Comments on the Quality of English Language

English: typos and sentences for which changes are highlighted in Yellow were suggested. It is not a complete list.  I would recommend revision by a native English scientist.

Author Response

The review discusses the diversity in virulence and immune responses against foot-and-mouth disease virus (FMDV) in susceptible host species, to serve as a foundation for a rational design of improved vaccines and countermeasure strategies, ideally with the aim to eradicate FMD.

The review is well written and informative. As a minor flaw, it tends to take for granted that the reader has at least basic knowledge of details of FMDV. In addition, it lacks figures. A figure of the NCR, or even of the whole genome, with its salient portions highlighted would be useful. I had to look one up online to help follow.

Major

Lines 88-89. a brief explanation of what the S segment is would be helpful.

Line 163-165. Were the pigs infected at all? Why is this interesting? Was the virus not infectious any more?

Lines 214-219 should be at the beginning of the paragraph to introduce protein 2c.

Line 277 (and Table 1) please explain what DIVA markers are, at least the acronym.

MINOR Comments

Table 1. I would suggest to add a column just for references rather than placing them after the improvement strategy. The journal style would be a number.

Line 94 attenuating should be attenuated

Line 101 Lpro should be introduced here

Line 135 impact on….

Line 151 “the infectious clone“ name is missing.

Line 159 effective should be introduced effectively.

Line 160 a virus should be a viral mutant.

Line 171 vaccine efficacy should be introduced efficacy as vaccines.

Line 172 this virus: which?

Line 320 In fact should be indeed.

Line 349 high  should be highly.

Line 368 should species not be serotype? Or are the authors referring to animal species?

Line 391 similar should be similarly.

Line 443 …known for their potent APCs should be …known for being potent APCs.

Line 456 Although, as in other DCs populations, FMDV infection of LCs is abortive. Meaning is not clear

Line 623 ..was removed… do the Authors imply that therefore its use is not restricted? Pls explain why this is an advantage.

We appreciate the reviewer for the positive feedback. In response to the reviewer's suggestion, we have developed a new figure illustrating the FMDV viral genome to enhance clarity for readers when discussing various viral proteins or non-translated regions. .Below, you will find a detailed list of major and minor comments that have been addressed individually

Mayor

The S fragment is and RNA structure within the 5’NCR of 350 nt that forms a stem loop. We have slightly modified the sentence to make it more clear for the reader.

The Leaderless virus when used as Life Attenuated Vaccine fails to induce protective immunity because it is too attenuated. Therefore, it does not have the opportunity to replicate a very low levels to induce immunity. However, it can still be used as the antigen for the Inactivated Vaccine and, since it is so attenuated, it is a more secure platform for antigen production. We have modified the sentence to make it more clear for the reader.

We have changed as suggested to introduce 2C accordingly.

DIVA acronym has been introduced in the text earlier. We have added to the table.

Minor comments

Table 1 modified as suggested by the reviewer.

Previous Line 94 the virus is “attenuated” but it has “attenuating” characteristics.

Previous Line 101 Lpro addressed.

Previous Line 135 impact on….We could not understand what the reviewer wanted to addressed.

Previous Line 151 “the infectious clone“ name has been added.

Previous Line 159 we do not agree with the change suggested.

Previous Line 160 we have mutant virus in the text.

Previous Line 171 “vaccine efficacy” has been addressed.

Previous Line 172 this virus: which? Leaderless, addressed.

Previous Line 320 Indeed added instead of In fact.

Previous Line 349 high should be highly. Addressed.

Previous Line 368 should species not be serotype? Or are the authors referring to animal species? It is referring to the animal host. We have modified the sentence to make it more clear for the reader.

Previous Line 391 similar should be similarly. Addressed.

Previous Line 443 …known for their potent APCs should be …known for being potent APCs. Addressed.

Previous Line 456 Although, as in other DCs populations, FMDV infection of LCs is abortive. Meaning is not clear. Sentence modified to make it clearer.

Previous Line 623 ..was removed… do the Authors imply that therefore its use is not restricted? Pls explain why this is an advantage. Because it reduces the requirements of policies and procedures designed to maintain the security. This has been added to the text.